# Ares: Predictable Traffic Engineering under Controller Failures in SD-WANs

## ABSTRACT

Emerging web applications (*e.g.*, video streaming and Web of Things applications) account for a large share of traffic in Wide Area Networks (WANs) and provide traffic with various Quality of Service (QoS) requirements. Software-Defined Wide Area Networks (SD-WANs) offer a promising opportunity to enhance the performance of Traffic Engineering (TE), which aims to enable differentiable QoS for numerous web applications. Nevertheless, SD-WANs are managed by controllers, and unpredictable controller failures may undermine flexible network management. Switches previously controlled by the failed controllers may become offline, and flows traversing these offline switches lose the path programmability to route flows on available forwarding paths. Thus, these offline flows cannot be routed/rerouted on previous paths to accommodate potential traffic variations, leading to severe TE performance degradation. Existing recovery solutions reassign offline switches to other active controllers to recover the degraded path programmability but fail to promise good TE performance since higher path programmability does not necessarily guarantee satisfactory TE performance. In this paper, we propose Ares to provide predictable TE performance under controller failures. We formulate an optimization problem to maintain predictable TE performance by jointly considering fine-grained flow-controller reassignment using P4 Runtime and flow rerouting and propose Ares to efficiently solve this problem. Extensive simulation results demonstrate that our problem formulation exhibits comparable load balancing performance to optimal TE solution without controller failures, and the proposed Ares significantly improves average load balancing performance by up to 43.36% with low computation time compared with existing solutions.

## KEYWORDS

Traffic Engineering, Software-Defined Wide Area Networks, Web Services, Controller Failures.

## 1 INTRODUCTION

Popular modern cloud services bring emerging new web applications (*e.g.*, web services [31], video streaming [10], web AR/VR [15], and tactile Internet [27]), which could be deployed for tenants various Quality of Service (QoS) requirements. These web services account for a large share of Wide Area Networks (WANs) traffic [38]. Traffic Engineering (TE) is a prevalent network application that aims to improve the network performance of WANs and enable differentiable QoS for numerous web applications [36]. TE plays a crucial role in the network operations of Internet Service Providers (ISPs) as it allows them to efficiently manage traffic distribution across their WANs. As Software-Defined Networking (SDN) has been introduced into WANs, also known as Software-Defined Wide Area Networks (SD-WANs) [39, 42], the management of WANs for

TE becomes much more flexible. Empowered by the SDN, TE can be implemented at the SDN controller, enabling it to respond promptly to traffic changes by leveraging a global network view [24]. Once TE generates an updated routing strategy based on measured Traffic Matrices (TMs) [45], the SDN controller can implement corresponding routing policies at the underlying SDN switches to reroute flows accordingly. Extensive evaluation results on WANs of world-leading giant techs (*e.g.*, Google [18, 20], Microsoft [17, 22], and Amazon [19]) have proven the effectiveness of achieving a better TE performance by introducing SDN into WANs.

While the SDN offers numerous network management benefits, it still faces challenges. The flexible management of network flows in an SD-WAN relies on the functionality of the SDN controller. However, controller failure is a common problem in SD-WANs [7, 16]. An SDN controller, typically a physical server or virtual machine running a network operating system, can experience unexpected failures due to various unforeseen circumstances (*e.g.*, power outage [40] and malicious attacks [21]). When a controller fails in an SD-WAN, all the switches previously controlled by the failed controller become offline switches. Any flow traversing these offline switches becomes offline and loses its path programmability, which denotes the ability to adjust the flow's forwarding path, and consequently, the controller can no longer control the flow. Based on our new observation in Section 2.2, when controller failures happen [11, 14], network performance cannot be guaranteed to accommodate potential traffic variations due to the loss of flexible network management. In some severe cases, several controllers may fail simultaneously or successively, leading to significant TE performance variation with a large number of offline flows.

To mitigate the impact of controller failures on web services due to TE performance degradation, state-of-the-art solutions attempt to reassign offline switches to other active controllers [8, 9, 12]. These solutions are typically path programmability-centric and take maximizing the total recovered path programmability as their primary objective. However, these path programmability-centric solutions may not necessarily guarantee better TE performance, since they fail to achieve satisfactory and predictable TE performance after controller failures occur, which is attributed to path programmability being unable to reveal network performance. While higher path programmability implies a higher likelihood of accommodating unpredictable future traffic fluctuations, it cannot guarantee a predictable TE performance when the current traffic status is given (*i.e.*, TMs are known). Thus, an effective recovery solution is needed to maintain web services by ensuring improved predictable TE performance under controller failures.

In this paper, we propose the *PredictAble TRaffic Engineering Solution (Ares)*, which aims to guarantee predictable TE performance under controller failures. The key idea of Ares is to jointly consider fine-grained flow-controller reassignment using P4 Runtime and flow rerouting based on the input TMs. We formulate an optimization problem named *TE Performance-aware Flow-Controller*

*Reassignment and Flow Rerouting (TPFCRFR)*, which seeks to minimize the Maximum Link Utilization (MLU) in the network under controller failures. The formulated TPFCRFR problem is a Mixed Integer Linear Programming (MILP) with high complexity. To tackle this problem, we propose an efficient and effective heuristic solution named Ares to determine the reassignment and rerouting policies. Extensive evaluation results verify the effectiveness of our formulation: the optimal solution of our formulation exhibits comparable load balancing performance to the optimal TE solution without controller failures. The results also show that the proposed Ares can improve the average load balancing performance by up to 43.36% with low computation time compared with existing solutions.

The main contributions of this paper are summarized as follows:

- We identify that controller failures may be a severe factor affecting web service performance due to TE performance degradation, and existing recovery solutions cannot guarantee predictable TE performance.
- We formulate an optimization problem to maintain predictable TE performance by jointly considering flow-controller reassignment and flow rerouting. We further conduct simulations to prove the effectiveness of our proposed TPFCRFR problem.
- To solve the problem, we propose Ares. Extensive simulations under real-world topology and traffic traces demonstrate that Ares can guarantee good TE performance with low computation time.

The remainder of this paper is organized as follows. In Section 2, we provide the background, observations, and motivation of this paper. Section 3 discusses the design opportunity and design overview of Ares. Section 4 formulates the TPFCRFR problem. In Section 5, we propose Ares to efficiently address this problem. In Section 6, we verify the effectiveness of our formulation and compare the performance of Ares with other existing baseline solutions. Section 7 discusses the related work within this research area. Finally, we conclude the paper in Section 8.

## 2 BACKGROUND AND MOTIVATION

In this section, we will introduce the background, motivation, and our new observations of this paper.

### 2.1 TE Meets Various QoS Requirements for Web Services through SD-WANs

Emerging new web services require various QoS requirements. These web services are highly prioritized in the whole network and account for a large share of traffic in WANs [38]. To improve the network performance to enable differentiable QoS, TE offers a promising solution to this issue. By Leveraging the capabilities of SDN, TE can be efficiently deployed at the control plane, enabling rapid response to traffic fluctuations in WANs and promise required QoS for web services through a global network perspective. Specifically, TE aims to find effective routing strategies that redistribute the traffic across the network, thereby alleviating network congestion and optimizing network performance [6, 36]. Based on the given network topology and traffic demands of flows, TE will formulate an optimization problem with a specific objective function, *e.g.*, minimizing the MLU, to decide the suitable path for each flow

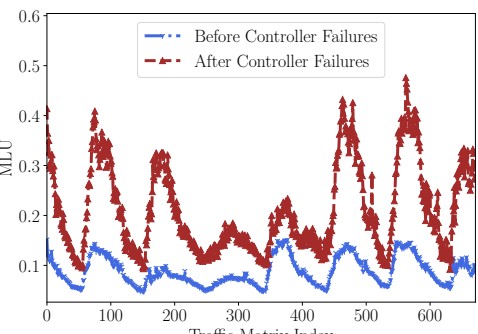

**Figure 1: Comparison of the MLU performance before and after controller failures. The lower, the better.**

to forward on. Since the SDN controller is able to maintain a global network view and update routing policies at the underlying SDN switches, TE will periodically reroute flows to achieve improved load balancing in response to dynamic changes in network traffic conditions for SD-WANs.

### 2.2 Controller Failures Pose Severe Impact on TE Performance

Even though SDN brings many benefits to WANs, it is still faced with challenges. SDN controllers, typically physical servers or virtual machines running a network operating system, may fail in SD-WANs. Unforeseen circumstances (*e.g.*, natural disasters, hardware or software errors, and power outages) can cause controller failures. When such controller failures occur, switches previously controlled by the failed controllers will become offline, and flows traversing these offline switches will lose their path programmability and become offline flows. The controller failure problem can cause fluctuations in network performance and even TE performance under varying network conditions due to decreased path programmability. To maintain stable network performance, it is crucial to recover these offline flows under controller failures.

Figure 1 shows our observation about the impact of controller failures on TE performance. The load balancing results are obtained by solving the Multi-Commodity Flow (MCF) problem [25], which is detailed in Appendix A.1. MCF is a widely used TE solution that balances traffic load among all links by minimizing the MLU. We utilize GÉANT topology with real-world TMs of the 23 switches at a time slot of every fifteen minutes and select 672 TMs collected during one week, specifically from June 2, 2005, to June 8, 2005. The blue line shows the MLU performance before controller failures, and the red one shows the MLU performance after controller failures. This observation demonstrates that controller failures threaten TE performance, which can increase the MLU by up to 0.35 in the worst case. The root cause lies in that flows cannot be flexibly rerouted to accommodate traffic fluctuations due to controller failures. Thus, meeting various QoS requirements for emerging web services under controller failures is challenging and non-trivial.

### 2.3 State-of-the-Art Solutions and Their Limitations

A common approach to the controller failure problem is to employ a cluster deployment strategy. The cluster deployment involves

setting up a controller cluster comprising multiple controllers to handle single controller failure [2, 3]. Typically, if one controller in the cluster fails, the other functioning controllers can continue operating and ensure uninterrupted control over the network. However, it is essential to note that if all controllers in a cluster fail simultaneously, such as during a power outage, this cluster design may not be effective [14]. Therefore, it is necessary to develop efficient and effective solutions to address this situation, which aims to maintain network performance to keep the network functional and responsive even under multiple controller failure scenarios.

To tackle this issue, existing state-of-the-art solutions propose reassigning offline switches to active controllers to maximize recovered path programmability under controller failures. However, higher path programmability can only indicate higher flexibility in network management and may not necessarily ensure the expected improvement in network performance, particularly in TE performance. In addition, existing solutions all follow the same coarse-grained switch-controller reassignment pattern, which limits the recovery performance since some flows with large traffic volumes may not be recovered for further rerouting. In other words, state-of-the-art can not promise good TE performance under controller failures. In the forthcoming Section 3, we will elaborate on the design of our proposed Ares, and how it overcomes these limitations.

## 3 DESIGN OVERVIEW OF ARES

In this section, we will introduce the opportunity to help us realize fine-grained flow-controller reassignment and present the design overview of Ares.

### 3.1 Opportunity

The development of the programmable data plane has facilitated precise flow-controller assignment without requiring additional hardware. P4 is a programming language utilized to program the data plane, as indicated by [5]. Additionally, P4 Runtime is a novel approach that allows the control plane to manage the data plane [4]. This specification is designed for P4 programmable switches and allows multiple controllers to manage the switch via the P4 Runtime server simultaneously. This feature aligns with the design goals of enabling multiple controllers to manage a single switch. By deploying a module in each P4 Runtime server, flow redirection from failed controllers to active ones can be accomplished using reassignment strategies. The reassignment strategies are computed by solving an optimization problem outlined in Section 4, and P4 Runtime allows for flow-controller reassignment among active controllers based on these strategies.

### 3.2 Overview of Ares

Ares aims to promise predictable TE performance under controller failures by jointly considering flow-controller reassignment and flow rerouting. As depicted in Figure 2, Ares has four main steps. Firstly, it collects real-time traffic traces (e.g., TMs) from the network periodically. Secondly, when controller failures occur, Ares updates the current network status with the required information (e.g., collected TMs, offline flows, and active controllers). Subsequently, flow-controller reassignment and flow rerouting policies

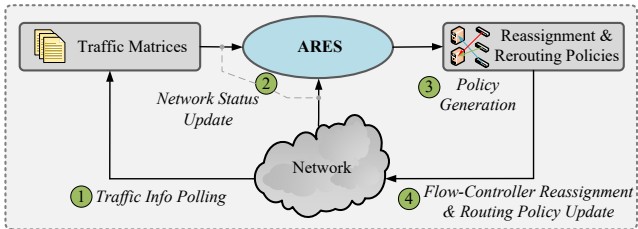

Figure 2: Design structure of Ares.

are determined and generated by solving the optimization problem (i.e., TPFCRFR problem) detailed in the following Section 4. Finally, Ares reassigns offline flows to corresponding active controllers and updates routing policies, achieving predictable load balancing performance in the whole network.

## 4 PROBLEM FORMULATION

In this section, we will exhibit the formulation of the TPFCRFR problem, which involves deciding the reassignment strategies for offline flows to active controllers under controller failures by considering the TE performance.

### 4.1 System Description

Typically, an SD-WAN consists of $H$ controllers at $H$ locations, and each controller controls a domain of switches. Controllers $C_{N+1}$, ..., $C_H$ fail, and the set of active controllers is $C = \{C_1, ..., C_i, ..., C_N\}$. The set of flows is $\mathcal{F} = \{f_1, ..., f_j, ..., f_T\}$, where $f_1, ..., f_M$ are offline flows and $f_{M+1}, ..., f_T$ are online flows controlled by active controllers. The set of links is $\mathcal{E} = \{e_1, ..., e_k, ..., e_K\}$. The pre-configured path set for flow $f_j$ is $\mathcal{P}_j = \{p_j^1, ..., p_j^l, ..., p_j^L\}$. We use $x_{ij}^l = 1$ to denote that offline flow $f_j$ ($j \in [1, M]$) is forwarded on path $p_j^l$ and reassigned to controller $C_i$; Otherwise $x_{ij}^l = 0$. We use $y_j^l = 1$ to denote that online flow $f_j$ ($j \in [M + 1, T]$) is forwarded on path $p_j^l$; Otherwise $y_j^l = 0$.

### 4.2 Problem Constraints

#### 4.2.1 Path Selection Constraint. Each offline flow must be forwarded on one path. Thus, we have:

$$\sum_{l=1}^{L} \sum_{i=1}^{N} x_{ij}^l = 1, \forall j \in [1, M]. \tag{1}$$

Each online flow should also select one path to forward on, which is:

$$\sum_{l=1}^{L} y_j^l = 1, \forall j \in [M + 1, T]. \tag{2}$$

#### 4.2.2 Flow-Controller Reassignment Constraint. When controller failures happen, active controllers must prioritize the management of flows from offline switches while ensuring uninterrupted normal operations. The control load of a controller is determined by the overall overhead involved in controlling the flows within its designated domain. We quantify a controller's control resource based on the number of flows it can effectively handle without introducing additional delays (e.g., queueing delay [41]).

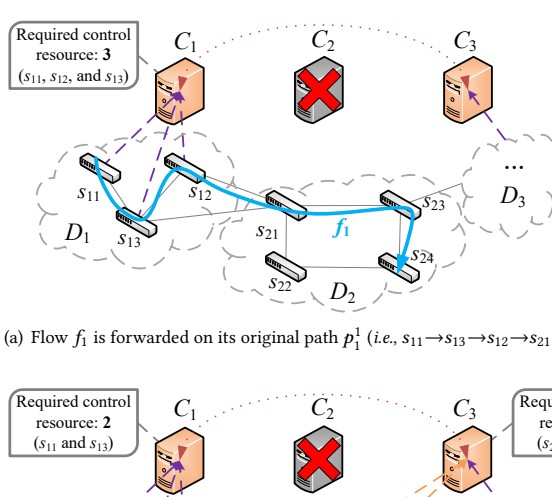

(a) Flow $f_1$ is forwarded on its original path $p_1^1$ (i.e., $s_{11} \rightarrow s_{13} \rightarrow s_{12} \rightarrow s_{21} \rightarrow s_{23} \rightarrow s_{24}$).

(b) To avoid the congested links $s_{12}$-$s_{21}$ and $s_{21}$-$s_{23}$, we have to reroute flow $f_1$ to a new path $p_1^2$ (i.e., $s_{11} \rightarrow s_{13} \rightarrow s_{21} \rightarrow s_{22} \rightarrow s_{24}$). For flow $f_1$, switches $s_{21}$ and $s_{22}$ have to be reassigned to another active controller to update corresponding flow entries. In this example, these two switches are reassigned to controller $C_3$. After the rerouting of flow $f_1$, the control load of controllers $C_1$ and $C_3$ has changed. Thus, we can calculate $h_1^2$ as 2, which refers to the increased control load of controller $C_3$. Furthermore, $g_{1,1}^2$ equals 1 since the available control resource of controller $C_1$ has been released for 1 unit due to the rerouting.

**Figure 3: An example to show how to calculate $h_j^l$ and $g_{ij}^l$. Controller $C_2$ fails, and all switches within domain $D_2$ become offline.**

In the worst case, the cascading failure may happen due to the overloading of controllers [43]. Hence, ensuring that a controller's control load remains within its available control resource is crucial. We use $h_j^l$ to denote the required available control resource of selecting path $p_j^l$ for flow $f_j$ and dispatching the control of flow $f_j$ to another active controller. We use $g_{ij}^l$ to denote the released control resource of controller $C_i$ if flow $f_j$ is forwarded on path $p_j^l$. Figure 3 shows an example to calculate $h_j^l$ and $g_{ij}^l$. Mathematically, the flow-controller reassignment constraint can be expressed as follows:

$$\sum_{l=1}^{L} \sum_{j=1}^{M} (x_{ij}^l * h_j^l) \le R_i^{avail.} + \sum_{l=1}^{L} \sum_{j=1}^{M} (x_{ij}^l * g_{ij}^l), \forall i \in [1, N], \quad (3)$$

where $R_i^{avail.}$ denotes the available resource of controller $C_i$.

*4.2.3 **Link Load Constraint**.* The total traffic load on each link, consisting of two parts (i.e., the sum of the traffic load from offline and online flows on each link), should be at most its upper bound link capacity. We use $Cap_k$ to denote link $e_k$'s capacity, $V_j$ to denote the traffic demand from flow $f_j$, $\alpha_k^l$ to denote the relationship

between links and paths, and $u$ to denote the MLU. The sum of the traffic load on link $e_k$ is denoted as $load_k$ and calculated as follows:

$$load_k = \sum_{l=1}^{L} \sum_{j=1}^{M} \sum_{i=1}^{N} (x_{ij}^l * V_j * \alpha_k^l) + \sum_{l=1}^{L} \sum_{j=M+1}^{T} (y_j^l * V_j * \alpha_k^l), \forall k \in [1, K].$$
(4)

Then, the traffic load on each link cannot exceed the link's capacity, and the link load constraint can be formulated as follows:

$$load_k \le Cap_k * u, \forall k \in [1, K]. \quad (5)$$

## 4.3 Objective Function

The objective of our proposed TPFCRFR problem is to minimize the MLU. Thus, the objective function can be formulated as follows:

$$obj = u.$$

## 4.4 Problem Formulation

Based on the above problem constraints and objective function, the problem can be formulated as follows:

$$\begin{aligned} \min_{u,x,y} \quad & u \\ \text{s.t.} \quad & (1)(2)(3)(4)(5), \\ & x_{ij}^l, y_j^l \in \{0, 1\}, \end{aligned} \quad (P)$$

where $\{x_{ij}^l\}$ and $\{y_j^l\}$ are binary design variables, and $u$ is a continues design variable. $\{Cap_k\}$ are given continues constants, $\{R_i^{avail.}\}$ are given integer constants, $\{\alpha_k^l\}$ are given binary constants, and $\{h_j^l\}$ and $\{g_{ij}^l\}$ are given integer constants. Given that the objective function is linear, this problem is a MILP.

## 5 SOLUTION

The typical approach to address the above TPFCRFR problem involves obtaining an optimal result through optimization solvers. However, the TPFCRFR problem is with high complexity. As the network grows, the solution space expands substantially, making finding a feasible solution within a reasonable time frame challenging or rendering it infeasible. Consequently, we propose a heuristic algorithm to solve the problem, aiming to provide a trade-off between performance and time complexity. The key idea is to select a path for each offline flow to forward on based on the probabilities obtained from the linear programming relaxation of problem (P). The details of the algorithm are summarized in Algorithm 1, and the notations used are listed in Table 1.

The algorithm begins by initializing an empty set $X$ (line 1). We first relax the binary variable $x_{ij}^l$ in problem (P) to continuous variables and obtain the linear programming relaxation solution $\bar{X}^*$. We then sort the values in $\bar{X}^*$ in descending order to get vectors $\bar{X}$ (line 2). Sorting the values in $\bar{X}^*$ in descending order helps prioritize all the tests based on their probabilities. From lines 4 to 17, the algorithm tests all possible selection and reassignment strategies by rounding the decimal values in $\bar{X}$ to configure proper reassignments. It first finds corresponding controller, flow, and path IDs $i_0$ and $j_0$ from $l_0$ (line 5). Then, it checks if the flow $f_{j_0}$ has not been reassigned yet and this reassignment is feasible based on problem constraints Eqs. (1) and (3). If it is a feasible one, this reassignment policy will be

**Table 1: Notations.**

| Notation | Meaning |
|----------|---------|
| $C$ | the set of active controllers, $C = \{C_i \mid i \in [1, N]\}$ |
| $\mathcal{F}'$ | the set of offline flows, $\mathcal{F} = \{f_j \mid j \in [1, M]\}$ |
| $\mathcal{P}'$ | the set of path sets for offline flows, $\mathcal{P} = \{P_j \mid j \in [1, M]\}$, where $\mathcal{P}_j = \{p_j^l \mid j \in [1, M], l \in [1, L]\}$ denotes all available paths for offline flow $f_j$ |
| $\mathcal{R}$ | the set of active controllers' available control resource, $\mathcal{R} = \{R_i^{avail.} \mid i \in [1, N]\}$ |
| $h_j^l$ | an integer constant that denotes required available control resource of selecting path $p_j^l$ for flow $f_j$ and dispatching the control of flow $f_j$ to an another active controller |
| $g_{ij}^l$ | an integer constant that denotes released control resource for controller $C_j$ if flow $f^j$ is forwarded on path $p_j^l$ |
| $\mathcal{X}$ | the set of the feasible reassignment relationship between active controllers, offline flows, and paths, $\mathcal{X} = \{(i, j, l) \in [1, N] \times [1, M] \times [1, L] \mid x_{ij}^l = 1\}$ |
| $\bar{X}$ | the set of testing mappings by solving the LP relaxation of problem (P) and sorting the results in the descending order, $\bar{X} = \{x_t, t \in [1, N * M * L]\}$ |

confirmed in $\mathcal{X}$, the available control resource of active controller $C_{j_0}$ will be updated, and the reassigned flow $f_{j_0}$ will be removed from the set $\mathcal{F}'$ (lines 7-9). Besides, the released control resource will also be updated to the corresponding controllers due to the selection of different paths for the flow.

If all offline flows have been reassigned, the first step of the algorithm will stop. As for the second step of this algorithm, if there remain unreassigned flows after the initial part, the algorithm proceeds to design a reassignment policy for these flows further. From lines 19 to 25, the algorithm lets each unreassigned flow $f_{j_0}$ keep forward on its original path $p_{j_0}^{l_0}$, which will not cause any further updates on active controllers' available control resource. The reassignment policy will be confirmed in $\mathcal{X}$, and the flow $f_{j_0}$ will be removed from the set $\mathcal{F}'$ (lines 22 to 23). Finally, in line 26, the algorithm stops, and the updated set $\mathcal{X}$ is returned.

## 6 EVALUATION

### 6.1 Evaluation Setup

In our performance evaluation of Ares, we utilize a representative backbone topology known as GÉANT. This topology is based on a network infrastructure in Europe and consists of 23 switches connected by 72 links. Each topology node has a unique ID associated with latitude and longitude coordinates. We deploy five SDN controllers for our evaluation setup, each with a control resource of 500 [8, 9, 12]. The control resource of a controller, as defined in previous studies [33, 35], refers to its processing capability to perform flow state pulling operations and retrieve network state variations without introducing additional control latency [41]. To conduct our evaluation, we utilize a dataset that records the real-world TMs of the 23 switches at a time slot of every fifteen minutes over four months [34]. From this dataset, we select 672 TMs collected during

---

**Algorithm 1:** Heuristic Algorithm

**Input** : $C, \mathcal{F}', \mathcal{P}', \mathcal{R}, h_j^l, g_{ij}^l$;

**Output:** $\mathcal{X}$;

1   $\mathcal{X} = \emptyset$;

2   generate $\bar{X} = \{x_t, t \in [1, N * M * L]\}$ by solving the linear programming relaxation of problem (P) and sorting the results in the descending order;

3   // select path for each offline flow to forward on and dispatch the control of the flow to an active controller based on the descending order of their probabilities;

4   **for** $x_{t_0} \in \bar{X}$ **do**

5     get controller, flow, and path IDs $i_0$, $j_0$, and $l_0$ from $x_{t_0}$;

6     **if** $\mathcal{X} \cup (i_0, j_0, l_0)$ satisfies constraints Eqs. (1) and (3) and $f_{j_0} \in \mathcal{F}'$ **then**

7       $\mathcal{X} \leftarrow \mathcal{X} \cup (i_0, j_0, l_0)$;

8       $\mathcal{F}' \leftarrow \mathcal{F}' \setminus f_{j_0}$;

9       $R_{i_0}^{avail.} = R_{i_0}^{avail.} - h_{j_0}^{l_0}$;

10      **for** $C_{i_0'} \in C$ **do**

11        $R_{i_0'}^{avail.} = R_{i_0'}^{avail.} + g_{i_0' j_0}^{l_0}$;

12      **end**

13     **end**

14     **if** $\mathcal{F}' == \emptyset$ **then**

15       break;

16     **end**

17   **end**

18   // test any remaining offline flows if there are any;

19   **if** $\mathcal{F}' \neq \emptyset$ **then**

20     **for** $f_{j_0} \in \mathcal{F}'$ **do**

21       get flow $f_{j_0}$'s original forwarding path $p_{j_0}^{l_0}$;

22       $\mathcal{X} \leftarrow \mathcal{X} \cup (*, j_0, l_0)$;

23       $\mathcal{F}' \leftarrow \mathcal{F}' \setminus f_{j_0}$;

24     **end**

25   **end**

26   **return** $\mathcal{X}$;

---

one week, specifically from June 2, 2005, to June 8, 2005, to serve as our dataset for evaluation purposes.

As for the pre-configured path set for MCF, OPT-TPFCRFR, and Ares, we utilize the pre-configured SMORE path set in our evaluation. SMORE is a widely recognized TE solution that effectively mitigates network congestion and achieves load balancing by intelligently selecting paths with dynamic weight adaptation [23]. The pre-configured SMORE path set is calculated using Räcke's oblivious routing algorithm [28]. Notably, SMORE leverages diverse paths to enhance network robustness and is optimized for load balancing. For our evaluation under the GÉANT topology, we select four SMORE paths with the highest weights for each flow. By leveraging SMORE's path selection algorithm, we aim to achieve efficient load balancing and improved performance in our evaluation.

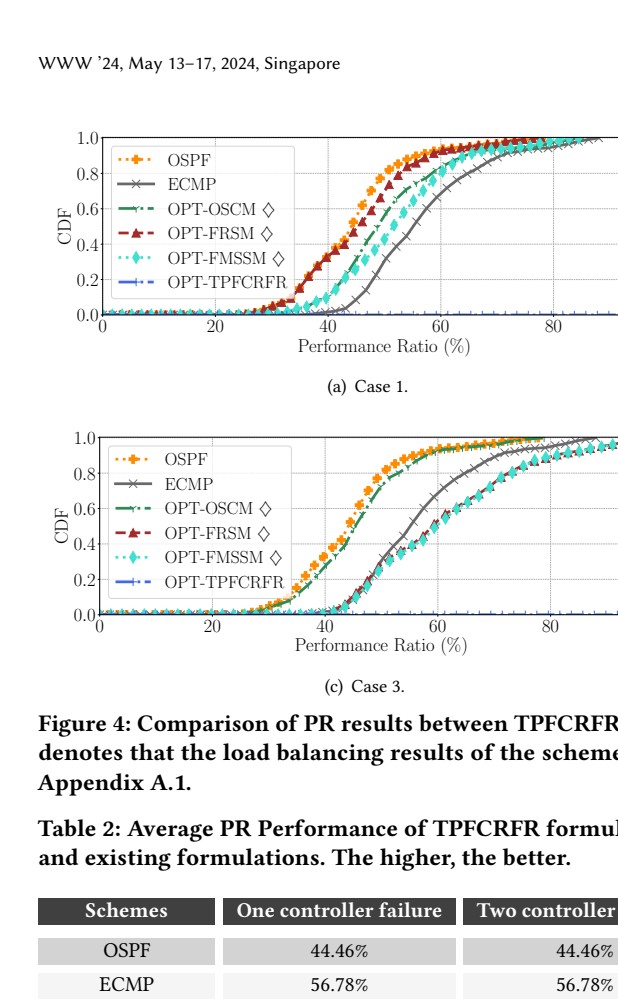

(a) Case 1.

(b) Case 2.

(c) Case 3.

(d) Case 4.

**Figure 4: Comparison of PR results between TPFCRFR formulation and existing formulations under two controller failures. ◇ denotes that the load balancing results of the scheme are obtained by using GUROBI to solve the MCF problem detailed in Appendix A.1.**

**Table 2: Average PR Performance of TPFCRFR formulation and existing formulations. The higher, the better.**

| Schemes | One controller failure | Two controller failures |
|---|---|---|
| OSPF | 44.46% | 44.46% |
| ECMP | 56.78% | 56.78% |
| OPT-OSCM [12]◇ | 60.57% | 55.68% |
| OPT-FRSM [9]◇ | 58.09% | 51.40% |
| OPT-FMSSM [8]◇ | 58.50% | 52.42% |
| OPT-TPFCRFR | 100.00% | 100.00% |

## 6.2 Comparison Algorithm

(1) Open Shortest Path First (**OSPF**) [26]: All flows are rerouted to their respective weighted shortest paths, which are calculated using the OSPF algorithm.

(2) Equal-Cost Multi-Path (**ECMP**) [32]: When multiple shortest paths exist between a pair of source and destination switches, the traffic is evenly distributed among all the available next hops along these paths. In other words, each switch along the path splits the traffic equally among all the corresponding shortest paths.

(3) Multi-Commodity Flow (**MCF**) [25]: This scheme formulates the MCF optimization problem based on the pre-configured path set, intending to minimize the MLU to achieve a decent load balancing performance. Details of the formulation can be found in Appendix A.1. We use GUROBI optimization solver [1] to solve this problem.

(4) **RetroFlow** [12]: This scheme aims to recover offline flows using a two-fold approach. Firstly, offline switches are configured to operate under the legacy routing mode. Secondly, the control of the rest of the offline switches is reassigned

to active controllers. OPT-OSCM is the optimal solution to the formulated problem in [12].

(5) **Matchmaker** [9]: This scheme adaptively modifies the control cost of offline switches based on the limited available control resource by intelligently rerouting flows to facilitate appropriate reassignment of offline switches. OPT-FRSM is the optimal solution to the formulated problem in [9].

(6) ProgrammabilityMedic (**PM**) [8]: This scheme focuses on recovering offline flows by ensuring that they have similar path programmability. It achieves this by determining the appropriate routing mode for flows at the recovered offline switches and configuring the mappings between switches and controllers. OPT-FMSSM is the optimal solution to the formulated problem in [8].

(7) Optimal results of the TPFCRFR problem (**OPT-TPFCRFR**): This scheme is the optimal solution to the formulated problem (P). We use GUROBI optimization solver [1] to solve this problem.

(8) **ARES**: This scheme is shown in Algorithm 1.

Note that the results presented for OSPF, ECMP, and MCF assume that no controller failures have occurred.

## 6.3 Evaluation Results

We evaluate the effectiveness of our proposed TPFCRFR formulation and ARES with the following two metrics. The first evaluation metric in our evaluation is the MLU. The lower MLU indicates a better TE performance. The second evaluation metric is the computation time. We want to ensure our proposed ARES is efficient and scalable enough to cope with severe controller failure scenarios and realize a fast recovery of the whole network.

To evaluate the load balancing performance, we employ the Performance Ratio (PR), which is defined as $PR = P_{\text{MCF}}/P_{\text{scheme}}$.

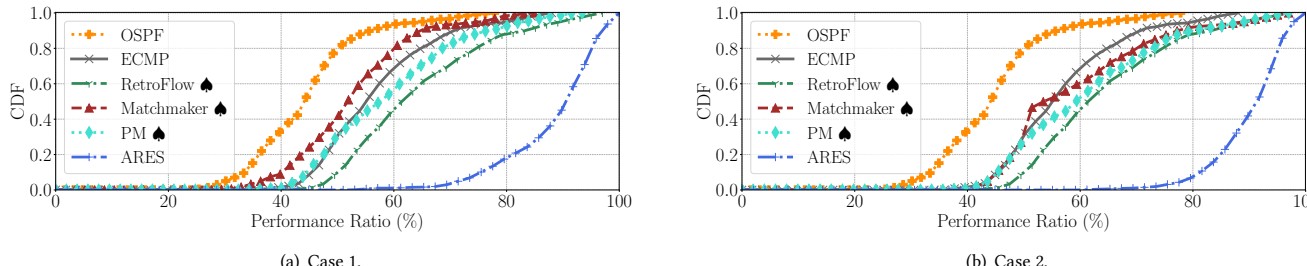

(a) Case 1.

(b) Case 2.

**Figure 5: Comparison of PR results between Ares and existing solutions under one controller failure. ♠ denotes that the load balancing results of the scheme are obtained by adopting the heuristic TE solution detailed in Appendix A.2 (Algorithm 2).**

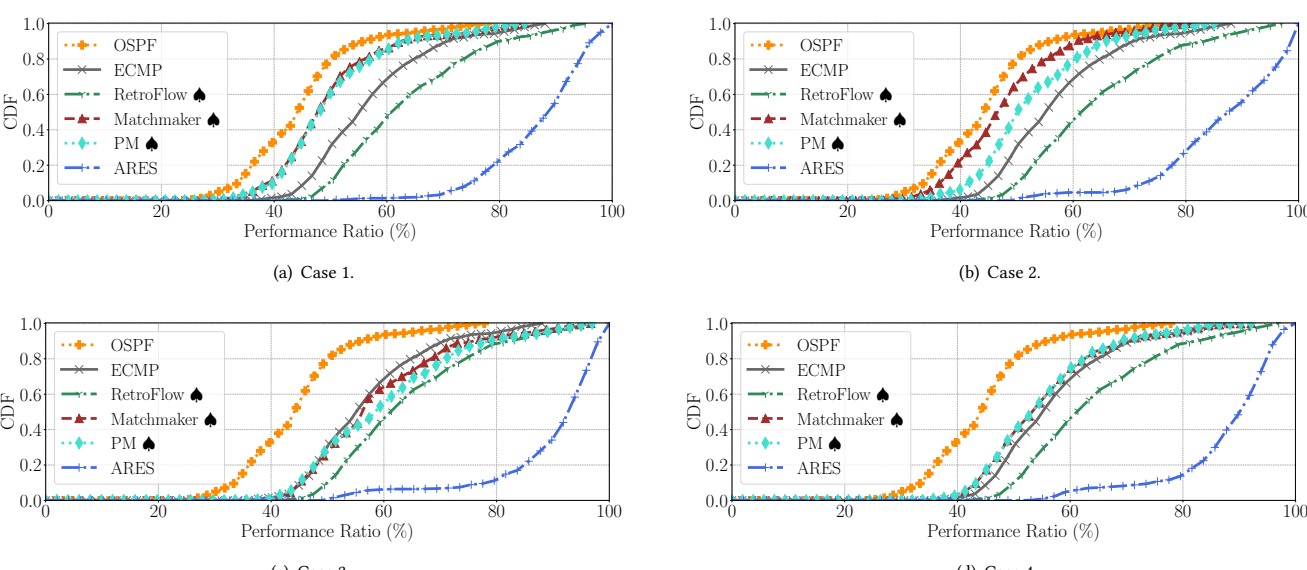

(a) Case 1.

(b) Case 2.

(c) Case 3.

(d) Case 4.

**Figure 6: Comparison of PR results between Ares and existing solutions under two controller failures. ♠ denotes that the load balancing results of the scheme are obtained by adopting the heuristic TE solution detailed in Appendix A.2 (Algorithm 2).**

$P_{\text{MCF}}$ represents the load balancing results achieved by the MCF approach without controller failures, which represents the optimal TE performance, and $P_{\text{scheme}}$ corresponds to the load balancing results attained by a specific scheme. A PR value of 1 implies that the scheme performs on par with the optimal results. A lower PR value indicates that the scheme's performance deviates significantly from the optimal results.

*1) Effectiveness of proposed TPFCRFR problem formulation.*
To evaluate the effectiveness of our formulated TPFCRFR formulation, we first compare the optimal results of the TPFCRFR problem with the optimal results of other problem formulations. Note that OPT-OSCM, OPT-FRSM, and OPT-FMSSM are the optimal results of the formulations for RetroFlow, Matchmaker, and PM, respectively. Given that these three schemes are only switch-controller reassignment solutions and do not include the TE operation part, we use ⋄ to denote that their load balancing results are obtained by using GUROBI to solve the MCF problem detailed in Appendix A.1.

Figure 4 shows four specific cases of PR results of TPFCRFR formulation and existing formulations under two controller failures.

It is important to note that when two out of five controllers fail, there are ten specific cases, but due to limited space, only four cases are displayed in this paper. We can see that OPT-TPFCRFR exhibits comparable TE performance with the optimal MCF solution and significantly outperforms other schemes. Table 2 shows the performance of optimal solutions' average PR. Compared with OSPF, OPT-TPFCRFR realizes the best TE performance and improves the load balancing performance by up to 55.54%. The simulation results prove that our TPFCRFR formulation benefits from jointly considering flow-controller reassignment and flow rerouting rather than separately doing the reassignment and rerouting in a two-step way in other schemes.

*2) Effectiveness of proposed Ares.* To further evaluate the effectiveness of our proposed Ares, we conduct more simulations on load balancing performance between our proposed Ares and other heuristics. Note that RetroFlow, Matchmaker, and PM are only switch-controller reassignment solutions and do not include the TE operation part. Thus, we use ♠ to denote that their load balancing results are obtained by adopting the heuristic TE solution

**Table 3: Average PR Performance of Ares and existing solutions. The higher, the better.**

| Schemes | One controller failure | Two controller failures |
|---|---|---|
| OSPF | 44.46% | 44.46% |
| ECMP | 56.78% | 56.78% |
| RetroFlow♠ | 64.01% | 63.70% |
| Matchmaker♠ | 57.46% | 50.49% |
| PM♠ | 60.97% | 51.44% |
| Ares | 87.82% | 87.23% |

**Table 4: Performance of average computation time to OPT-TPFCRFR. The lower, the better.**

| Schemes | One controller failure | Two controller failures |
|---|---|---|
| OPT-TPFCRFR | 100% | 100% |
| Ares | 2.53% | 2.07% |

detailed in Appendix A.2 (Algorithm 2) after the recovery procedure. Figures 5 and 6 illustrate six specific cases of heuristic solutions' PR results under one and two controller failures. From these cases, we observe that Ares consistently demonstrates compatibility with the optimal MCF performance and outperforms the other solutions. The evaluation results confirm that Ares effectively guarantees robust TE performance under controller failures since Ares jointly considers flow-controller reassignment and rerouting.

Table 3 presents the average PR of six solutions under both one and two controller failure scenarios. In the case of a single controller failure, Ares achieves the best performance compared to other schemes. Conversely, OSPF and ECMP exhibit the weakest performance as they cannot adapt the routing policy to dynamic traffic conditions. Compared to OSPF, Ares significantly improves load balancing performance, achieving up to a 43.36% enhancement. Additionally, RetroFlow, Matchmaker, and PM perform less effectively than Ares, indicating that path programmability-centric solutions fail to promise predictable TE performance under controller failures compared with Ares. Under two controller failures, the network faces more critical challenges, and maintaining satisfactory load balancing performance becomes crucial. Multiple controller failures can lead to more significant TE performance degradation due to an increased number of offline flows, and the performance of the solutions becomes vital. As illustrated in the table, Ares ensures robust TE performance and dramatically outperforms the rest of the solutions.

*3) Efficiency of proposed Ares.* We analyze the computation time for Ares and OPT-TPFCRFR in different scenarios and record the results in Table 4. On average, Ares only requires 2.53% and 2.07% of the computation time needed by OPT-TPFCRFR in the two scenarios mentioned above. As the network size increases and the failure scenario becomes more complex, the performance gap between OPT-TPFCRFR and Ares is expected to widen. Even though Ares has a slightly lower load balancing performance than OPT-TPFCRFR, it still proves to be a highly efficient solution with low computation complexity.

## 7 RELATED WORK

**Improving TE performance for web services.** Handigol *et al.* [13] present Plug-n-Serve, a load-balancing system for web services hosted in enterprise and campus networks that uses OpenFlow to control the routing of HTTP requests. The system aims to minimize the response time by taking both the congestion of the network and the load on the servers into account, and also propose to dynamically adjust the allocation of requests based on an integrated optimization algorithm called LOBUS. QoS-RL [44], a reinforcement learning-based TE solution, is proposed to provide good quality of service and load balancing for different priority levels of traffic in WANs. QoS-RL uses destination-based forwarding entries to reduce management overhead and service disruption, and also leverages reinforcement learning to intelligently select and update a few entries to reroute high and low-priority traffic concerning different objectives. Wang *et al.* [37] propose a joint optimization of TM measurement and TE for SDNs, considering the TCAM capacity and flow aggregation constraints. They formulate the joint optimization problem as a MILP model and propose two heuristic algorithms to design flow rules for traffic measurement and routing tasks. However, none of the above-mentioned solutions considers the impact of controller failures on TE performance and how to improve TE performance under controller failures further.

**Maintaining control resiliency in SD-WANs.** Ruchel *et al.* [30] evaluate the robustness of two open-source distributed SDN controllers, ONOS [2] and ODL [3], in different failure scenarios in both data and control planes. This paper measures the performance of the controllers in terms of latency, throughput, consistency, and recovery time using various topologies and traffic types, and also discusses the strengths and weaknesses of each controller and provides suggestions for future improvements. SDN-ESRC [29], a secure and resilient control plane for SDN based on endogenous security, uses multiple heterogeneous controllers to detect and correct malicious control messages. It also designs a scheduling algorithm to select the optimal controller set for each flow and presents a security evaluation model based on the Markov chain to analyze the performance of the proposed scheme under different attack scenarios. He *et al.* [14] propose a preventive priority setting model to balance the load among multiple controllers in SDNs. The model assigns a priority for each controller to become the primary controller of a switch, and automatically switches to the highest-priority available controller when a failure occurs. Nevertheless, all the above-mentioned solutions fail to consider improving TE performance, which means that they cannot provide decent network performance, especially for delay-sensitive web services.

## 8 CONCLUSION

In this paper, we propose Ares to promise predictable TE performance in SD-WANs during controller failures. Ares jointly considers the fine-grained flow-controller reassignment and flow rerouting in a single optimization problem and tries to minimize the MLU based on the given TMs. Extensive evaluation results show that not only our proposed TPFCRFR problem formulation exhibits comparable load balancing performance to optimal TE solution without controller failures but also brings significant improvements in load balancing performance with low computation time compared with existing solutions.

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

# A APPENDIX

## A.1 Formulation of the MCF Problem

In an SD-WAN, there are a total of $H$ switches and $K$ links connecting these switches. It is important to ensure that the utilization of each link, denoted as $e_k$ ($k \in [1, K]$), does not exceed its specified upper bound capacity, denoted as $Cap_k$. The set of flows is denoted as $\mathcal{F} = \{f_1, f_2, ..., f_j, ..., f_M\}$. Each flow $f_j$ in the set $\mathcal{F}$ has a pre-configured path-set consisting of $L$ paths, denoted as $\mathcal{P}_j = \{p_j^1, p_j^2, ..., p_j^l, ..., p_j^L\}$. The traffic demand for flow $f_j$ is denoted as $V_j$. We use $\alpha_k^l$ to denote the relationship between links and paths. Additionally, for each path $p_j^l$, the traffic demand routed on that particular path is represented by the binary variable $z_j^l$. The MLU is denoted as $u$. The objective of the optimization problem is to minimize the MLU (*i.e.*, $u$) in order to achieve optimal network performance while meeting the traffic demands of all flows. Therefore, the MCF formulation can be presented as follows:

$$\min_{u,z} \quad u$$

$$\text{s.t.} \quad \sum_{j=1}^{M} z_j^l = 1,$$

$$\text{(P-MCF)}$$

$$\sum_{l=1}^{L} \sum_{j=1}^{M} z_j^l * V_j * \alpha_k^l \le Cap_k * u,$$

$$z_r^l \in \{0, 1\}, \forall j \in [1, M], \forall l \in [1, L],$$

where $\{y_j^l\}$ is a binary design variable and $u$ is a continuous design variable, $\{V_j\}$, $\{Cap_k\}$, $\alpha_k^l$ are given constants.

## A.2 Heuristic Solution to the MCF Formulation

The key idea of the heuristic algorithm to the MCF formulation is similar to the proposed Algorithm 1, which aims to select a path for each flow to forward on based on the probabilities obtained from the linear programming relaxation of problem (P-MCF). The details of the algorithm are summarized in Algorithm 2. The algorithm begins by initializing an empty set $\mathcal{Z}$ (line 1). We first relax the binary variable $z_j^l$ in problem (P-MCF) to continuous variables and obtain the linear programming relaxation solution $\bar{Z}^*$. We then sort the values in $\bar{Z}^*$ in descending order to get vectors $\bar{Z}$ (line 2). Sorting the values in $\bar{Z}^*$ in descending order helps prioritize all the tests based on their probabilities. From lines 4 to 13, the algorithm tests all possible path selections by rounding the decimal values in $\bar{Z}$ to configure proper selections. It finds corresponding flow and path IDs $j_0$ from $l_0$ (line 5) and checks if the flow $f_{j_0}$ has not been selected yet. Then, the path selection will be confirmed in $\mathcal{Z}$ and the selected flow $f_{j_0}$ will be removed from the set $\mathcal{F}$ (lines 7-8). If all flows have been selected, the algorithm will stop (lines 10-12). Finally, in line 14, the updated set $\mathcal{Z}$ is returned.

---

**Algorithm 2:** Heuristic Algorithm to the MCF Formulation

**Input** : $\mathcal{F}$;
**Output** : $\mathcal{Z}$;

1   $\mathcal{Z} = \emptyset$;
2   generate $\bar{Z} = \{z_t, t \in [1, M * K]\}$ by solving the linear programming relaxation of problem (P-MCF) and sorting the results in the descending order;
3   // select path for each flow to forward on based on the descending order of their probabilities;
4   **for** $z_{t_0} \in \bar{Z}$ **do**
5     get flow and path IDs $j_0$ and $l_0$ from $z_{t_0}$;
6     **if** $f_{j_0} \in \mathcal{F}$ **then**
7       $\mathcal{Z} \leftarrow \mathcal{Z} \cup (j_0, l_0)$;
8       $\mathcal{F} \leftarrow \mathcal{F} \setminus f_{j_0}$;
9     **end**
10     **if** $\mathcal{F}'' == \emptyset$ **then**
11       break;
12     **end**
13   **end**
14   **return** $\mathcal{Z}$;

---

