# OpenReview forum: "ARES: Predictable Traffic Engineering under Controller Failures in SD-WANs"
_ACM.org/TheWebConf/2024/Conference — TheWebConf24_

### Official Review · Reviewer_8cQR · 2023-11-22

**Novelty:** 4
**Technical Quality:** 4

**Review:**

Review Comments:
In this paper, a solution is proposed for the scenario of controller failure, addressing both flow-controller reassignment and flow rerouting collaboratively. The authors tackle the routing problem by creating an MILP model and leveraging an optimizer to offline reassign flows to active controllers while rerouting them along alternative paths. Subsequently, through relaxation, the complexity of the problem and computation time are reduced. Extensive experiments demonstrate the superior performance of the proposed approach compared to other solutions in the context of controller failure.

Advantages:
The formalization of the controller failure issue and performance enhancement through optimization.
The relaxation of the original problem and the introduction of a heuristic algorithm to reduce computation time.

Disadvantages:
Experimentation is confined to the GEANT topology, and it is recommended to extend experiments to various network topologies.
Figures 4, 5, and 6 show the comparisons of only one
performance metrics. It is advisable to include comparisons of more diverse metrics, such as computation time in comparison to other algorithms.
A more detailed description of algorithm complexity and parameter count is necessary.
Some minor issues, such as the ambiguous statement "We use 𝛼 to denote the relationship between links and paths."

**Questions:**

When controller failure occurs, can we choose new paths through LP and then reassign offline flows to active controllers using a simple balancing algorithm?

Is there a specific reason preventing the reassignment of online flows along with offline flows to active controllers? It would be more convenient to reassign all flows through a heuristic method.

Since the paths of ARES are the same as those of the SMORE algorithm, why not compare ARES with SMORE?

**Reviewer Confidence:**

4: The reviewer is certain that the evaluation is correct and very familiar with the relevant literature

**Scope:**

4: The work is relevant to the Web and to the track, and is of broad interest to the community

---

### Official Review · Reviewer_84hR · 2023-11-23

**Novelty:** 6
**Technical Quality:** 5

**Review:**

This takes a fairly standard traffic engineering problem but formulates it as a problem of reassigning controllers. The paper is well written and mathematically appears sound. The evaluation is extremely painstaking and thorough. However, I found myself really struggling to understand the results in the experimental set up. It seems good work but the presentation made it painfully hard to understand in parts.

The traffic engineering design problem is formulated as minimising the proportion to which the most loaded link is loaded. This is a common assumption but may simply fail to be useful in some circumstances (e.g. some source of traffic with a single link has a load which is very close to capacity.)
They take a relatively standard approach of relaxing an integer problem to a continuous problem. The write up is clear and as far as I can tell mathmatically sound.

I find figure 4 very hard to understand. It talks about four cases which I can't find described anywhere. I've read the paper a few times so I apologise if I miss ths. Figure 5 only has two cases and figure 6 is back to four. It is not made at all clear why there is a CDF here. I guess they run the schemes for the specified time and compare the performance ratio over time to get a CDF but I can't find this at all made clear in the text so I do have to guess where the CDF originates in these results.

The text says P_MCF and P_scheme are the "performance" which from context I think is the MLU (as this is stated earlier).

In an early part of the problem formulation explicit time is given to describing the hazards of controller overload but this seems to be dropped as an issue by the evaluation section (or does it remain in constraint 3).

Grammar/typos:

4.4 "continues constants" "continues design variable" should be continuous.

5 "However, the TPFCRFR problem is with high complexity." poor grammar

**Questions:**

An assumption is made that managing a flow is one resource unit for a controller and this has no dependency on the number of packets in that flow or the complexity of the management performed. Is this a big limitation? It seems likely to be true if all we are putting in place is a routing rule per flow but SDN rules can be a lot more complex than that: imagine flow A requires a single new rule in the switch but flow B requires several.

We presumably want each switch assigned to one and only one controller? Is that the case? If so where is this handled in the constraints?

The text describes the table as giving the average PR for one and two controller failures (is it the average or median?) I think this corresponds somehow to the experiments in figure 4 but it is not clear which cases.

The text notes that the results presented for MCF "assume that no control failures have occurred" but the performance ratio is defined as the ratio of the performance of the scheme relative to P_MCF with no failures. Can you clarify this? (I can't actually see results for MCF anyway -- is the PR there 1 by definition?)

If table 3 is average performance over several runs can we see standard deviation as well to assess consistency?

Constraint 3 appears to be the only one about controller overload during assignment. It does not seem to be evaluated. It does not seem to ensure that switches are "all" assigned to a single controller. Could this be made more clear?

**Reviewer Confidence:**

3: The reviewer is confident but not certain that the evaluation is correct

**Scope:**

4: The work is relevant to the Web and to the track, and is of broad interest to the community

---

### Official Review · Reviewer_Dpqd · 2023-11-24

**Novelty:** 6
**Technical Quality:** 5

**Review:**

## Summary

Software-define networking is becoming increasingly popular as a solution to traffic engineering challenges in wide area networks. However, software-defined wide area networks (SD-WANs) may have their performance undermined by controller failures. When this happens, switches previously controlled by the failed controller essentially turn in to offline switches; that means that flows crossing those switches lose programmability, and so cannot be routed (or rerouted) on previous paths. While existing recovery mechanisms maximise programmability, this doesn't necessarily guarantee wider traffic engineering performance. This paper proposes "ARES", an algorithm that provides a solution to the optimisation problem of providing predictable traffic engineering performance in the face of controller failures. The proposed algorithm is shown to improve average load balancing performance by 43%, vs. existing solutions.

## Reasons to accept

- The paper is well-written, and presents a novel approach to a known problem in SDNs for wide-area networks.
- The optimisation problem is well specified.
- The proposed algorithm is comprehensively evaluated against a reasonable set of alternatives.

## Reasons to reject

- The evaluation is conducted by simulating the algorithm over traces taken from an existing network. It isn't clear how generalisable the results will be to other networks, or real-world conditions.

**Questions:**

- Can the generalisability of the results be discussed/quantified? The traces are taken from one network, but the paper doesn't describe how representative that is likely to be.

**Reviewer Confidence:**

2: The reviewer is willing to defend the evaluation, but it is likely that the reviewer did not understand parts of the paper

**Scope:**

3: The work is somewhat relevant to the Web and to the track, and is of narrow interest to a sub-community

---

### Official Review · Reviewer_b2WH · 2023-11-27

**Novelty:** 4
**Technical Quality:** 4

**Review:**

Pros:
This manuscript proposes an algorithm ARES to promise predictable TE performance in SD-WANs during controller failures. Specifically, ARES jointly considers the flow-controller reassignment and the flow rerouting, with the goal of minimizing the MLU.

Cons:
a) Motivation needs to be strengthened. The authors state that the coarse-grained switch-controller reassignment pattern in existing work may lead to unsuccessful flow recovery, particularly for flows with large traffic volumes, thus limiting the overall recovery performance. Consequently, this factor is covered by ARES for better network performance. Nevertheless, this fundamental statement lacks convincing evidence, particularly in term of data-driven empirical results.
b) Algorithm 1 requires to be refined. The authors do not fully analyze either the lower/upper bound or the time complexity of the proposed heuristic solution. These two aspects are essential to evaluate the algorithm’s effectiveness and practicality
c) Evaluation requires to be improved. The authors only evaluate the performance of ARES under limited numbers of controller failures (i.e., one and two failures). Would ARES still maintain its improvement when considerable failures, such as half of controllers, happen?
d) Novelty needs justification, given many existing studies such as “an efficient online algorithm for dynamic SDN controller assignment in data center networks”, ton’17, etc.

**Questions:**

a) Motivation needs to be strengthened. The authors state that the coarse-grained switch-controller reassignment pattern in existing work may lead to unsuccessful flow recovery, particularly for flows with large traffic volumes, thus limiting the overall recovery performance. Consequently, this factor is covered by ARES for better network performance. Nevertheless, this fundamental statement lacks convincing evidence, particularly in term of data-driven empirical results.

b) Algorithm 1 requires to be refined. The authors do not fully analyze either the lower/upper bound or the time complexity of the proposed heuristic solution. These two aspects are essential to evaluate the algorithm’s effectiveness and practicality.

c) Evaluation requires to be improved. The authors only evaluate the performance of ARES under limited numbers of controller failures (i.e., one and two failures). Would ARES still maintain its improvement when considerable failures, such as half of controllers, happen?

**Reviewer Confidence:**

2: The reviewer is willing to defend the evaluation, but it is likely that the reviewer did not understand parts of the paper

**Scope:**

3: The work is somewhat relevant to the Web and to the track, and is of narrow interest to a sub-community

---

### Official Review · Reviewer_zbG3 · 2023-12-04

**Novelty:** 3
**Technical Quality:** 4

**Review:**

Paper Summary:
The paper addresses the challenge of maintaining Traffic Engineering (TE) performance in Software-Defined
Wide Area Networks (SD-WANs) during controller failures. It proposes a novel solution that jointly considers
fine-grained flow-controller reassignment and flow rerouting, aimed at minimizing the Maximum Link Utilization
(MLU) in these networks. The effectiveness is demonstrated through simulations under real-world topology
and traffic traces, showing improvements in load balancing and efficiency compared to existing solutions.

Strengths:
● Detailed Problem Formulation: The problem constraints and formulations are detailed, reflecting a deep
understanding of the complexities in network management.

● Comprehensive Comparative Analysis: The paper includes a thorough comparative analysis with
various existing algorithms, which provides a robust context for assessing ARES performance.

● Robust Methodology: The methodology, particularly the use of a Mixed Integer Linear Programming
(MILP) model and a heuristic algorithm, is well thought out, balancing theoretical rigor with practical
applicability.

Weaknesses:
● Motivation: The paper does not describe the motivation clearly, especially in the production SD-WAN.

● Scalability Concerns: The paper does not provide an extensive analysis of the scalability of ARES, which
is crucial for understanding its applicability in larger and more complex network environments.

● Limited Scope in Evaluation: While the GÉANT topology is used, the evaluation might lack diversity in
network scenarios and topologies, potentially limiting the generalizability of the results.

I truly tried to like this paper given the prospects in SD-WAN, but I think it is simply not ready for prime time. Let
me elaborate in the question section below.

**Questions:**

In a production SD-WAN, each controller would have multiple instances synced with each other by consensus
algorithm. The probability seems small that all instances controlling the same set of resources are down at the
same time. Thus, the motivation for the paper could be justified.
Moreover, once the controller fails or the control plane loses connection to the switches, the switch would
enter a massive fail open state (keep whatever is programmed as it is). The global controller can then be
notified of this failure, and recalculates the traffic tunnel rules. Would this be more realistic, compared with
detecting the failure, and asking another controller to take over.

The paper employs a Mixed Integer Linear Programming (MILP) model to formulate the TE problem under
controller failures. The solution proposed is a heuristic algorithm. Besides the existing constraints, are there
any additional constraints to consider in a production SD-WAN setting, e.g., network scalability, QoS
requirements, security considerations, etc.?

Is the GÉANT Topology general enough for the evaluation? While the GÉANT topology provides a realistic
setting, incorporating additional network topologies would validate the effectiveness of ARES in varied network
environments.

Regarding the controller failure scenarios, does it make difference if it is a simultaneous failures among
controllers, sequential failures, and failures under peak load conditions

Would the traffic pattern affect the evaluation results? e.g., evaluating ARES with dynamic and unpredictable
traffic patterns to gauge its real-world robustness.

Scalability is also one of my concerns. How would the ARES scale in larger networks with more controllers and
switches to understand its performance in enterprise or carrier-grade environments. It would be great if
showing the computation time and resource utilization scale with network size.

The paper has limited discussion on integration with existing SD-WAN systems. There are several production
SD-WAN systems deployed in the industrial production network. The overall system architecture has been
discussed in the corresponding papers. It would be appearing to integrate or interoperate with existing traffic
engineering solutions and network management practices.

**Reviewer Confidence:**

3: The reviewer is confident but not certain that the evaluation is correct

**Scope:**

3: The work is somewhat relevant to the Web and to the track, and is of narrow interest to a sub-community

---

### Decision · Program_Chairs · 2024-01-22

**Decision:**

Accept

**Comment:**

This paper presents a mechanism to cope with controller failures in a software-defined wide-area network in order to perform flow controller reassignment and flow rerouting, modelled a Mixed Integer Linear Programming (MILP) plus a heuristic algorithm.

 This paper sees two classes of reviews concerning novelty and technical strength, featuring lower and higher ratings. Looking at the reviews and discussions, I am split. On the one hand, the authors manage to show that their design could also work with multiple network topologies (as was raised as a generalization question by multiple reviewers). On the other hand, there is a notable concern about the practicality and implementability of a distributed contoller and its responsibility for individual switches, as particularly the discussion with reviewer 84hR shows. In part, the authors and the reviewer talked past each other and I got the same questions and concerns about the authors points raised on a geo-distributed controller. This appears questionable, at least non-obvious (and the author comments rather handwaving) as any distributed consensus yields further latency. The second issue that the authors could not resolve is that of "non-unit resources" when it comes to rules in one or spread across multiple flow tables. Here, it appears that the authors' persistent references to paper from 2013 and 2014 suggest an underlying switch model that may be not primarily relevant anymore.

 Leaving aside some of the practicalities, there does not seem to be anything fundamentally wrong with the paper, but it did not spur much excitement either.

 This makes it rather a borderline paper for me.